# De-Escalation of Anti-Tumor Necrosis Factor Alpha Agents and Reduction in Adverse Effects: A Systematic Review

**DOI:** 10.3390/biomedicines10051034

**Published:** 2022-04-29

**Authors:** Marleen Bouhuys, Willem S. Lexmond, Patrick F. van Rheenen

**Affiliations:** Department of Pediatric Gastroenterology, Hepatology and Nutrition, University of Groningen, University Medical Centre Groningen, Beatrix Children’s Hospital, 9713 GZ Groningen, The Netherlands; m.bouhuys@umcg.nl (M.B.); w.s.lexmond@umcg.nl (W.S.L.)

**Keywords:** tumor necrosis factor inhibitors, infliximab, adalimumab, etanercept, golimumab, certolizumab pegol, dose reduction, interval lengthening, adverse effects

## Abstract

Background: The long-term use of anti-TNF-α agents can lead to adverse effects, such as infections and immune-mediated cutaneous reactions. Whether de-escalation by dose reduction or interval lengthening reduces these adverse effects is uncertain. This systematic review aims to compare the incidence of infections and skin manifestations after anti-TNF-α dose de-escalation with standard dosing. Methods: MEDLINE, EMBASE, and the Cochrane Central Register of Controlled Trials were searched from inception to 14 January 2022. Randomized controlled trials (RCTs) and observational studies comparing anti-TNF-α de-escalation strategies with standard dosing among patients with inflammatory conditions, that report on infections, skin manifestations, or both, were included. The risk of bias was assessed with the revised Cochrane risk-of bias tool (RCTs) or the Newcastle–Ottawa scale (non-RCTs). Results: Fourteen RCTs and six observational studies (or 2706 patients) were included. Eight RCTs had low risk of bias or some concerns. Four non-RCTs were of good methodological quality. The studies described patients with axial spondyloarthritis (8 studies, 780 patients), rheumatoid arthritis (7 studies, 1458 patients), psoriasis (3 studies, 332 patients), or inflammatory bowel disease (2 studies, 136 patients). De-escalation strategies included interval lengthening (12 studies, 1317 patients), dose reduction (6 studies, 1130 patients), or both (2 studies, 259 patients). Overall, the occurrence of infections and skin manifestations did not differ between standard treatment and de-escalation. The disappearance of infections or skin manifestations after de-escalation was only reported in two studies. The majority of studies focused on etanercept and adalimumab. Heterogeneity in reporting of infections and skin manifestations precluded meta-analysis. Conclusion: We found that anti-TNF-α de-escalation does not reduce infections or skin reactions. A de-escalation strategy should not be recommended for the sole purpose of reducing drug-related adverse effects. The meticulous documentation of adverse effects is recommended to further address this question. Registration: PROSPERO CRD42021252977.

## 1. Introduction

Anti-tumor necrosis factor alpha (TNF-α) agents are used in a variety of inflammatory diseases, including inflammatory bowel diseases (IBD), rheumatic diseases (rheumatoid arthritis (RA) and axial spondyloarthritis), and psoriasis [1]. Antibodies targeting TNF-α inactivate the proinflammatory effect by direct neutralization. As a result, the TNF-α-dependent cytokine cascade is interrupted, which leads to the downregulation of inflammatory pathways. Adalimumab, infliximab, certolizumab pegol, golimumab, and etanercept all target the same epitope and are successfully used in the treatment of one or more of the above-mentioned inflammatory disorders [1,2].

Because of the effectiveness of anti-TNF-α agents, patient outcomes have greatly improved, and sustained remission has become a realistic treatment target [3,4,5,6,7]. The downside of long-term exposure to anti-TNF-α agents is its association with adverse effects, such as immune-mediated cutaneous reactions and susceptibility to infections.

In a recent meta-analysis including patients with IBD treated with anti-TNF-α agents, the pooled incidence of any dermatological reaction was 19% (95% confidence interval (CI) 15–24), with psoriasis or psoriasiform rash being the most common [8]. The pathophysiology of this paradoxical adverse effect is not completely clear; however, a local increase in interferon alpha (because its release is no longer inhibited by TNF-α) and genetic predisposition may play a role [9]. Other associated dermatological adverse effects include injection/infusion site reactions (caused by allergic hypersensitivity reactions and/or local trauma), eczema (possibly caused by an imbalance in the type 1 and 2 helper T cell response), and skin infections [8,9,10,11,12]. In a meta-analysis focusing on patients with RA, psoriatic arthritis, and ankylosing spondylitis, 30.8% of the anti-TNF-α users had at least one infection during follow-up. Compared to non-exposed patients, the odds ratio for any infectious adverse event was 1.20 (95% CI 1.10–1.30) [13]. Additionally, patients exposed to anti-TNF-α agents tend to require intravenous antibiotics or hospitalization and are at greater risk of developing tuberculosis [11,13,14,15].

The occurrence of adverse effects, along with high costs, frequent hospital visits, and injections, justify anti-TNF-α de-escalation studies. Dose de-escalation can be achieved by either reducing the dose of individual administrations or by lengthening the interval between drug administrations. The majority of de-escalation studies have been performed in patients with RA and axial spondyloarthritis. Multiple systematic reviews and meta-analyses of de-escalation studies demonstrated little or no increase in disease activity compared with standard dosing [16,17,18,19,20]. A Cochrane meta-analysis, including 3315 participants in total, showed that there was no loss of clinical response in about 80% of recent onset RA patients within the first year after dose reduction in etanercept. Additionally, anti-TNF-α dose reduction did not affect a proportion of patients in sustained remission (risk ratio 1.01, 95% CI 0.80–1.28) [16].

It is unknown whether dose de-escalation reduces adverse effects. We carried out a systematic review to evaluate the incidence of infections and skin manifestations after anti-TNF-α de-escalation, irrespective of underlying inflammatory conditions. The outcomes of interest included the disappearance or reduction in infections and skin manifestations, as well as the rate of occurrence of new infections and skin manifestations.

## 2. Methods

The current systematic review was performed according to the Preferred Reporting Items for Systematic Reviews and Meta-Analyses (PRISMA) statement and checklist [21,22]. This systematic review was included in the control group of the GoodReports randomized trial (GRReaT) [23]. The GRReaT team anonymously and independently assessedthe manuscript for completeness by reporting against the PRISMA checklist. Their feedback was incorporated in the manuscript before submission for publication. The completed PRISMA checklist can be found in Appendix A.

### 2.1. Data Sources and Searches

We searched MEDLINE (through PubMed), EMBASE, and the Cochrane Library from inception to 14 January 2022. The search strategies were developed in collaboration with a medical information specialist and consisted of a Boolean association of keywords, combining keywords for anti-TNF-α agents and de-escalation. The search strategies for each of the electronic databases are shown in Appendix A. All searches were carried out on 14 January 2022. No language restrictions were applied. In addition, we hand-searched references of relevant publications to identify any additional studies that were missed in the database searches.

### 2.2. Study Selection

To be included in this systematic review, studies had to be full-text articles based on randomized controlled trials (RCTs) or observational cohort or case–control studies involving patients treated with standard-dosed anti-TNF-α agents (adalimumab, infliximab, certolizumab pegol, golimumab, or etanercept), undergoing anti-TNF-α de-escalation (dose reduction or interval lengthening), and had to contain information about the rate of disappearance and/or reduction in infections and/or skin manifestations, as well as the rate of occurrence of new infections and/or skin manifestations. With the inclusion of patients, regardless of their underlying inflammatory condition, we expected to identify a sufficient number of studies to reach an acceptably high level of evidence. Studies in which de-escalation resulted in discontinuation without separate information about adverse effects during the de-escalation phase were excluded. Case reports, case series, and studies with a cross-sectional design were also excluded.

The search results were imported into EndNote (version 20, Clarivate, Philadelphia, PA, USA) for de-duplication [24], and subsequently imported into Rayyan, an online tool used for systematic reviews [25]. Two reviewers (M.B. and P.F.v.R.) independently reviewed titles and abstracts for eligibility. In case eligibility was unclear based on the title and abstract, the record was included for full-text assessment. Disagreements were solved by referral to a third reviewer (W.S.L.). Next, full-text articles were screened independently by the same two reviewers. Again, disagreements were solved by referral to the third reviewer. The study selection process was summarized in a flow diagram.

### 2.3. Data Extraction and Quality Assessment

Data were extracted by one reviewer (M.B.) and confirmed by another reviewer (P.F.v.R.). The following characteristics were extracted from each selected study: the first author, the year of publication, the report title, the name of the study, the corresponding author’s contact information, the country of origin, the publication type, the study design, the in- and exclusion criteria, the allocation method, anti-TNF-α dosing (standard dosing and de-escalation method), the start date, the end date, the duration of participation, the sample size, baseline imbalances, withdrawals and exclusions, the patient characteristics at baseline (age, sex, diagnosis, anti-TNF-α agent(s) used, co-treatment), incidence/prevalence and/or disappearance/reduction in infections and/or skin manifestation at baseline and at each reported time point, as well as the type and severity of the adverse reactions. Missing data were requested from the study authors via email. Missing and unobtainable data were presented as ‘not provided’ and were not included in the syntheses.

The risk of bias for the outcome of interest of RCTs was assessed with the revised Cochrane risk-of bias tool (RoB 2) in Microsoft Excel (version 2016, Microsoft Corporation, Redmond, WA, USA) [26]. The risk of bias was scored by two reviewers (M.B. and P.F.v.R.) as ‘low’, ‘some concerns’, or ‘high’ for each domain individually and overall. These results were displayed in a figure. The risk of bias of non-RCTs was assessed with the Newcastle–Ottawa quality assessment scale (NOS), converted to the Agency for Healthcare Research and Quality standards (‘good’, ‘fair’, or ‘poor’), and displayed in a table [27,28].

### 2.4. Data Analysis and Synthesis of Results

The occurrence of infections and skin manifestations was presented as the proportion of patients with at least one event for the standard treatment and the de-escalation group for each study individually. 95% CIs were calculated for these proportions using the Wilson method. Difference in these proportions were considered to be statistically significant if there was no overlap in the 95% confidence intervals of the two groups or if the *p*-value, if provided, was below 0.05. A relative difference of ≥25% was considered numerically different. The narrative syntheses were grouped by adverse event type. Results were summarized narratively and characteristics, and main results of included studies were presented in a table.

## 3. Results

In total, 2280 articles were identified, 128 of which were retrieved for full-text review. Of these, 108 were excluded as they did not report the outcome of interest (Figure 1). Table 1 lists the characteristics of 20 studies that compared any form of anti-TNF-α dose de-escalation with standard dosing [29,30,31,32,33,34,35,36,37,38,39,40,41,42,43,44,45,46,47,48]. Of these, 14 were RCTs (2197 patients; eight trials with low risk of bias or some concerns, as shown in Figure 2) [29,30,31,32,33,34,35,36,37,38,39,40,41]. Four of six non-RCTs (509 patients) were of good methodological quality (Table 2) [43,44,45,46,47,48].

Eight studies reported on axial spondyloarthritis (including ankylosing spondylitis) (780 patients) [31,33,35,38,39,40,42,43,44,45], seven reported on rheumatoid arthritis (1458 patients) [30,32,34,36,37,41], three reported on psoriasis (332 patients) [29,39,47], and two reported on inflammatory bowel disease (136 patients) [46,48].

Etanercept, which is usually administered subcutaneously once or twice a week, was evaluated in 14 studies [29,30,31,32,33,34,35,37,38,39,42,43,44,45]. Adalimumab, normally administered subcutaneously every other week, was evaluated in eight studies [37,38,39,41,44,46,47,48]. Certolizumab, which is usually administered subcutaneously every other week until remission and then every four weeks, was evaluated in two studies [36,40]. The intravenous administration of infliximab, which is usually carried out with three induction doses over 6 weeks (weeks 0–2–6), followed by maintenance therapy every 8 weeks, was evaluated in two studies [38,44]. Golimumab, commonly administered subcutaneously every four weeks, was evaluated in one study [38]. In Appendix A, the standard regimen and the dose de-escalation strategy for each study are described in detail.

### 3.1. Occurrence of New Infections

Seventeen of twenty articles reported on the occurrence of new infections during the study observation period [29,30,31,32,33,34,35,36,38,39,40,41,42,43,44,45,47,49].

#### 3.1.1. Any Infection

Eleven studies reported the occurrence of any infection [32,33,34,36,38,39,41,42,43,44,47]. There was no unequivocal evidence in favor of either de-escalation or standard therapy.

The RCT by Raffeiner et al., which included 323 patients with RA, reported a higher incidence rate in the standard treatment group, as compared to the de-escalated group (17.2 per 100 person years (PYs) versus 10.4 per 100 PYs, respectively; *p* < 0.001) [34]. The risk of bias of this study was high due to missing data and a lack of information about the method of data collection.

Two studies reported numerically fewer new infections in de-escalated patients (defined as a relative group difference ≥25%); however, this difference did not reach statistical significance. The REDES-TNF trial reported an incidence of 30.6% (95% CI 20.6–43.0) in patients on standard treatment and 18.0% (95% CI 10.4–29.5 vs. 30.6%) in de-escalated patients [38]. The cohort study by Závada et al. reported an incidence of 10.8% (95% CI 5.8–19.3) in patients on standard treatment and 7.5% (95% CI 3.0–17.9) in de-escalated patients [44].

In contrast with the former two studies, the ANSWERS trial, a small RCT with a high risk of bias, reported numerically more infections in de-escalated patients (43.5%, 95% CI 25.6–63.2 vs. 33.3%, 95% CI 17.9–53.3, requested data) [33].

Five other studies reported no differences in the occurrence of new infections between standard treatment and de-escalation (i.e., relative difference <25%) [36,39,41,42,43,47]. Among them was one RCT with a low risk of bias (C-EARLY-2). In this study, a lengthened certolizumab pegol interval (200 mg every 4 weeks, *n* = 127) was compared to standard treatment (200 mg every 2 weeks, *n* = 83) in patients with RA and sustained low disease activity. After 52 weeks of follow-up, 38.6% (95% CI 30.6–47.3) of the patients in the de-escalated group had at least one infection compared to 31.3% (95% CI 22.4–41.9) in the standard treatment group [36].

#### 3.1.2. Serious Infections

Eight studies reported the occurrence of serious or severe infections [30,32,34,35,36,41,42,45]; however, a clear definition was only given in two [30,34]. In six of these studies, no or few patients developed a serious infection, and there was no difference in the incidence rates between patients on standard treatment and de-escalated patients [32,35,36,41,42,45]. The PRESERVE study found higher incidences of treatment-emergent serious infections in the standard treatment group compared to the de-escalation group (1.5%, 95% CI 0.5–4.3 vs. 0.0%, 95% CI 0.0–1.9); however, this was not statistically significant [30]. On the contrary, the RCT by Raffeiner et al. found more severe infections in the de-escalation group (incidence rate 0.67 vs. 0.23 per 100 person years); however, this was also not significant [34].

#### 3.1.3. Specific Infections

Five studies reported on specific infections, including upper respiratory tract infections, flu, urinary tract infections, oral candidiasis, and tuberculosis [29,30,31,40,42,45]. None of the studies found significant differences between anti-TNF-α standard treatment and de-escalation. Neither was a difference in the type of infections reported between standard treatment and de-escalation. Most infections were mild and opportunistic infections were uncommon [29,30,31,32,38,39,43,45,47,49].

### 3.2. Disappearance of Infections

Only one article included information about the disappearance of infections after anti-TNF-α de-escalation. In this retrospective case–control study by van Steenbergen et al., frequent infectious symptoms disappeared in five out of seven (71.4%, 95% CI 26.2–69.0) IBD patients that underwent adalimumab interval lengthening from 40 mg every other week to every three weeks. In the standard treatment group, a disappearance occurred in none of the patients with frequent infectious symptoms at baseline (*n* = 5) [48].

### 3.3. Occurrence of New Skin Manifestations

Eleven articles included information about the occurrence of new skin manifestations after anti-TNF-α de-escalation [29,30,31,33,35,36,37,39,43,45,47]. No study described a difference in the type of skin manifestations between standard treatment and de-escalation.

#### 3.3.1. Any Skin Manifestation

Five studies reported on the occurrence of any skin manifestation [33,36,37,39,47]. The C-EARLY-2 trial, an RCT with a low risk of bias (C-EARLY-2), found numerically higher incidences of skin manifestations in the standard treatment group compared to the de-escalation group (9.6%, 95% CI 5.0–17.9 vs. 7.1%, 95% CI 3.8–12.9); however, this difference was not statistically significant [36]. The CONDOR study, a somewhat smaller RCT, showed a similar trend with more skin manifestations in the standard treatment group (7.3%, 95% CI 2.5–19.4 vs. 5.1%, 95% CI 1.4–16.9) (requested data) [39]. During the open-label extension of this RCT, none of the patients with a lengthened interval had a new skin manifestation, compared to 12.5% in the standard treatment group (requested data) [47]. In the ANSWERS trial, patients that underwent an Etanercept dose reduction experienced more new skin manifestations than patients on standard treatment; however, this was not statistically significant, and the study had a high risk of bias [33]. In the OPTTIRA study that only reported the number of events instead of the number of incidences, a skin manifestation occurred 13 times in the standard treatment group (*n* = 19, no serious events) and 24 times in the de-escalated patient group (*n* = 44, two serious events) [37].

#### 3.3.2. Injection Site Reactions

Injection site reactions were reported in six studies [29,31,33,35,43,45]. Four studies looked at the incidence of etanercept-related injection site reactions with and without a dose reduction [29,33,43,45]. Two of them reported significantly fewer injection site reactions in de-escalated patients. In the cohort study by Park et al., the incidence rate ratio was significantly smaller in de-escalated patients compared to patients on standard treatment (0.327, 95% CI 0.134–0.801, *p* = 0.014) [43]. The RCT by Papp et al. reported injection site reactions in 18.0% (95% CI 13.3–24.1) of the patients on standard treatment vs. 3.7% (95% CI 1.8–7.4) of the de-escalated patients [29]. However, we considered its 12-week follow-up (only 8 weeks after de-escalation) to be insufficient to draw meaningful conclusions.

Two studies reported numerically (but not significantly) fewer injection site reactions in de-escalated patients [31,45] and one study found no differences [33].

Surprisingly, the RCT by Li et al. reported more injection site reactions in patients that underwent interval lengthening (11.5% vs. 0.0%, not significant); however, in this study, follow-up after de-escalation also lasted for only 8 weeks [35].

### 3.4. Disappearance of Skin Manifestations

Two articles included information about the disappearance of skin manifestations after anti-TNF-α de-escalation. In the previously mentioned case–control study, de-escalation resulted in the disappearance of skin manifestations in 47.1% of the patients with skin manifestations at baseline. The disappearance of psoriasiform lesions was most common (two out of four, 50.0%), followed by xerosis cutis (three out of seven, 42.8%). No skin manifestations disappeared in the standard treatment group [48].

In a retrospective cohort study by Pouillon et al., skin manifestations linked to anti-TNF-α therapy disappeared in four out of seven patients (57.1%, 95% CI 25.1–84.2) after adalimumab interval lengthening [46]. However, based on the Newcastle–Ottawa scale, this study was of poor methodological quality, mainly because of the absence of a control group.

## 4. Discussion

In this systematic review, we summarize results from 20 anti-TNF-α de-escalation studies (14 RCTs and 6 non-RCTS) on the reduction in infections and skin manifestation. We considered two de-escalation strategies (dose reduction and interval lengthening) and included all underlying inflammatory conditions and all currently available anti-TNF-α agents.

After synthesizing the data, we conclude that anti-TNF-α de-escalation in patients treated according to the label does not reduce the occurrence of infections and skin manifestation compared to patients who continued standard dosing. However, the quality of evidence is low. It is unclear whether anti-TNF-α de-escalation improves existing infections and/or skin abnormalities.

There are multiple reasons why both patients and healthcare professionals may wish to de-escalate anti-TNF-α therapy once remission is achieved. These reasons include a reduction in the number of hospital visits, the number of needle pricks, and costs. A reduction in anti-TNF-α-associated adverse effects is also often mentioned as a reason; however, we cannot confirm the validity of this approach. We suggest not to de-escalate standard-dosed anti-TNF-α medication solely for this reason. This advice does not apply to patients treated with a shorter dosing interval, a higher dose than standard, or both.

Our findings are consistent with a meta-analysis performed by Vinson et al. They evaluated the incidence of serious infections and adverse events of specific interest in patients with RA or axial spondyloarthritis. Thirteen studies were included in the meta-analysis, seven of which were also included in our systematic review. The de-escalation of the biological disease-modifying anti-rheumatic drug (predominantly anti-TNF-α agents) the or the JAK inhibitor was not different from continuation of the initial regimen with respect to the incidence of serious infections (risk difference 0.01, 95% CI −0.00–0.02, *p* = 0.13, I^2^ = 0%). In contrast to our systematic review, Vinson et al. did not study dermatological adverse effects or non-serious infections [50].

Likewise, a Cochrane systematic review on the down-titration and discontinuation of anti-TNF-α agents in patients with RA also did not report on the occurrence of infections or skin manifestations. Based on five studies, four of which are also included in our systematic review, the authors concluded that de-escalation has little to no effect on serious adverse effects; however, the evidence was also very uncertain [16].

Although eight RCTs and four non-RCTs were of good overall methodological quality in our systematic review, measuring the outcome of interest was problematic. None of the studies defined adverse effects as their primary outcome. Instead, data about infections or skin manifestations were at best presented as part of the obligatory reporting of adverse events. As a consequence, most studies were not powered to detect potential differences in the occurrence of adverse events. Additionally, clear descriptions of the definitions and methods of measurement were lacking.

In most studies, the occurrence of new infections was expressed as incidences. This may have resulted in an underestimation of the effect of anti-TNF-α de-escalation on adverse effects. For instance, if a patient in the standard treatment group had 10 infections during follow-up and another patient in the de-escalation group had only 1, different incidences cannot be attained, whereas the use of event rates can provide a more reliable estimate.

We only included publications that reported on infections, skin manifestations, or both. Because of the obligatory reporting of adverse events, it is likely that these data were also available for de-escalation studies that did not provide this information in their publication. In fact, 106 studies were excluded during the full-text selection because they contained no or insufficient information about the adverse events of interest. This may have caused selection bias; however, obtaining these missing data from such a large number of publications is not feasible.

We included patients regardless of their underlying inflammatory condition and synthesized the data as if they were from one group. We anticipated to pool the data and perform subgroup analyses for each diagnosis and anti-TNF-α agent; however, due to limited data and heterogeneity, no valid results could be generated. Further research is necessary for better quality data on the possible beneficial effect of anti-TNF-α de-escalation on anti-TNF-α-associated adverse effects. This is of particular importance for patients with IBD and psoriasis and the anti-TNF-α infliximab, certolizumab, and golimumab agents, which were underrepresented in the current review. To better address the question whether de-escalation reduces adverse effects, future studies should scrupulously register the adverse effects of interest, by sending out questionnaires specifically designed for this purpose, for instance [51]. Another method to reduce exposure to anti-TNF-α agents and its associated adverse effects is the administration of this drug in cycles, with anti-TNF-α free periods in between. This concept is currently being investigated [52].

In conclusion, anti-TNF-α de-escalation does not seem to reduce infections or skin manifestations in patients on standard-dosed anti-TNF-α treatment; however, the available evidence is of low quality. We recommend against anti-TNF-α de-escalation for the sole purpose of putting an end to these adverse effects. Adequately powered studies with meticulous documentation of adverse effects are likely to increase the certainty of the evidence.

## Figures and Tables

**Figure 1 biomedicines-10-01034-f001:**
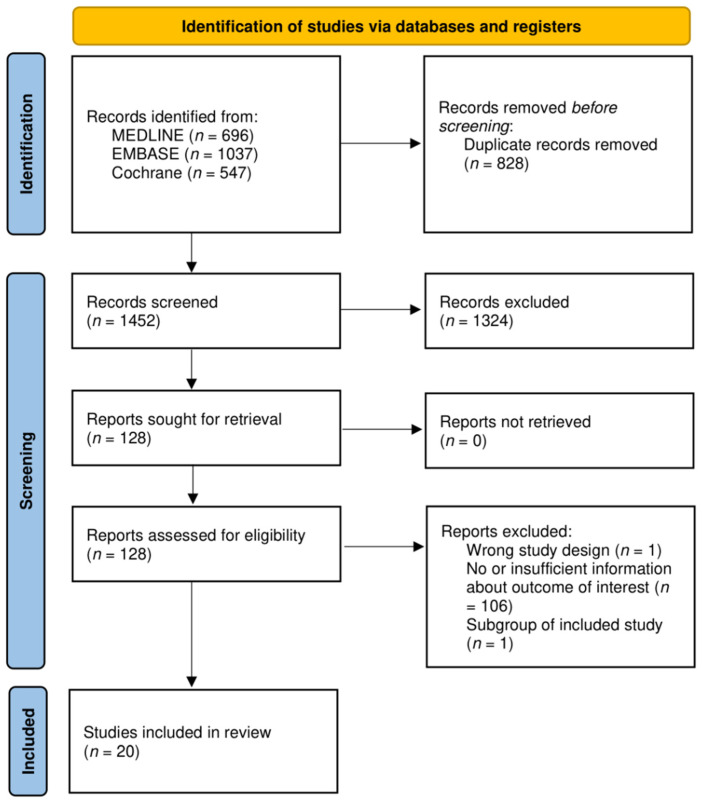
PRISMA 2020 flow diagram of study selection.

**Figure 2 biomedicines-10-01034-f002:**
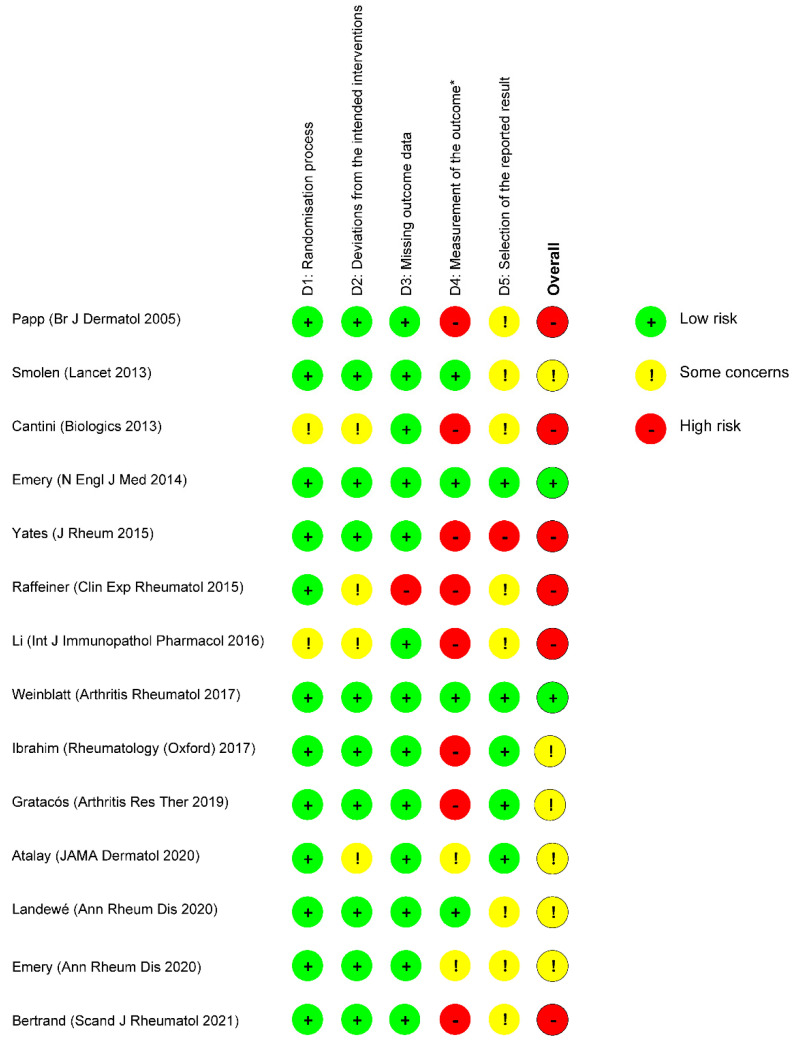
Risk of bias of RCTs (revised Cochrane risk-of bias tool). * Scored for outcome of interest of this systematic review (i.e., the occurrence of new infections and/or skin manifestations, or the reduction and/or disappearance of infections and/or skin manifestations).

**Table 1 biomedicines-10-01034-t001:** Characteristics and main results of included studies.

Study (Acronym)	Design	Number of Patients	Diagnosis	Anti-TNF-α Agent	De-Escalation Method	Duration of Follow-Up	Main Results Adverse Effects (*n* of Patients with ≥1 Event (%, 95% CI))
Papp (Br J Dermatol, 2005) [29]	RCT	194 (194 ST; 190 DE, subgroup in which all patients received ST followed by DE)	Psoriasis	Etanercept	Dose reduction	12 weeks	**Occurrence of new infections:** Upper respiratory tract infections: ST: *n* = 25 (12.9%, 8.9–18.3) DE: *n* = 24 (12.6%, 8.6–18.1) ‘Flu syndrome’: ST: *n* = 8 (4.1%, 2.1–7.9) DE: *n* = 5 (2.6%, 1.1–6.0) **Occurrence of new skin manifestations:** Injection site reactions: ST: *n* = 35 (18.0%, 13.3–24.1) DE: *n* = 7 (3.7%, 1.8–7.4)
Smolen (Lancet, 2013) [30] (PRESERVE)	RCT	404 (202 ST; 202 DE)	Rheumatoid arthritis	Etanercept	Dose reduction	52 weeks	**Occurrence of new infections:** Nasopharyngitis: ST: *n* = 17 (8.4%, 5.3–13.1) DE: *n* = 10 (5.0%, 2.7–8.9) Bronchitis: ST: *n* = 12 (5.9%, 3.4–10.1) DE: *n* = 11 (5.4%, 3.1–9.5) Treatment-emergent serious infections: ST: *n* = 3 (1.5%, 0.5–4.3) DE: *n* = 0 (0.0%, 0.0–1.9) **Occurrence of new skin manifestations:** Herpes Zoster: ST: *n* = 4 (2.0%, 0.8–5.0) DE: *n* = 1 (0.5%, 0.1–2.8) Malignant melanoma: ST: *n* = 1 (0.5%, 0.1–2.8) DE: *n* = 0 (0.0%, 0.0–1.9)
Cantini (Biologics, 2013) [31]	RCT	43 (21 ST; 22 DE)	Axial spondylo-arthritis	Etanercept	Interval lengthening	Mean of 22 months (SD 1.1)	**Occurrence of new infections:** Urinary tract infections: ST: *n* = 1 (4.8%, 0.8–22.7) DE: *n* = 2 (9.1%, 2.5–27.9) Upper respiratory tract infections: ST: *n* = 5 (23.8%, 10.6–45.1) DE: *n* = 7 (31.8%, 16.4–52.7) **Occurrence of new skin manifestations:** Injection site reactions: ST: *n* = 4 (19.0%, 7.7–40.0) DE: *n* = 3 (13.6%, 4.7–33.3)
Emery (N Engl J Med, 2014) [32] (PRIZE)	RCT	306 (306 ST; 63 DE, all patients received ST followed by DE in a subgroup of patients)	Rheumatoid arthritis	Etanercept	Dose reduction	ST: 52 weeks DE: 39 weeks	**Occurrence of new infections:** Any infection: ST: not provided DE: *n* = 17 (27.0%, 17.6–39.0) Serious infections: ST: *n* = 6 (2.0%, 0.9–4.2) DE: *n* = 1 (1.6%, 0.3–8.5)
Yates (J Rheum, 2015) [33] (ANSWERS)	RCT	47 (24 ST; 23 DE)	Axial spondylo-arthritis	Etanercept	Dose reduction	6 months	**Occurrence of new infections: ^1^** ST: *n* = 8 (33.3%, 17.9–53.3) DE: *n* = 10 (43.5%, 25.6–63.2) **Occurrence of new skin manifestations: ^1^** Any skin manifestation: ST: *n* = 3 (12.5%, 4.3–31.0) DE: *n* = 5 (21.7%, 9.7–41.9) Injection site reactions: ST: *n* = 2 (8.3%, 2.3–25.9) DE: *n* = 2 (8.7%, 2.4–26.8)
Raffeiner (Clin Exp Rheumatol, 2015) [34]	RCT	323 (164 ST; 159 DE)	Rheumatoid arthritis	Etanercept	Interval lengthening	Mean of 18 months (SD 1.2)	**Occurrence of new infections:** Any infection: incidence rate per 100 PYs ST: 17.2 DE: 10.4 *p* < 0.001 Severe infections: incidence rate per 100 PYs ST: 0.23 DE: 0.67 Not statistically significant
Li (Int J Immunopathol Pharmacol, 2016) [35]	RCT	43 (17 ST; 26 DE)	Axial spondylo-arthritis	Etanercept	Interval lengthening	12 weeks ^2^	**Occurrence of new infections:** Minor infections: ST: *n* = 1 (5.9%, 1.0–27.0) DE: *n* = 1 (3.8%, 0.7–18.9) **Occurrence of new skin manifestations:** Injection site reactions: ST: *n* = 0 (0.0%, 0.0–18.4) DE: *n* = 3 (11.5%, 4.0–29.0)
Weinblatt (Arthritis Rheumatol, 2017) [36] (C-EARLY-2)	RCT	210 (83 ST; 127 DE)	Rheumatoid arthritis	Certolizumab pegol	Interval lengthening	52 weeks	**Occurrence of new infections:** Any infection: ST: *n* = 26 (31.3%, 22.4–41.9) DE: *n* = 49 (38.6%, 30.6–47.3) Serious infections: ST: *n* = 1 (1.2%, 0.2–6.5) DE: *n* = 1 (0.8%, 0.1–4.3) Occurrence of new skin manifestations: ST: *n* = 8 (9.6%, 5.0–17.9) DE: *n* = 9 (7.1%, 3.8–12.9)
Ibrahim (Rheumatology (Oxford), 2017) [37] (OPTTIRA)	RCT	47 (19 ST; 44 DE, DE includes 16 patients from the ST group that were re-randomized to DE)	Rheumatoid arthritis	Adalimumab Etanercept	Interval lengthening	6 months	**Occurrence of new skin manifestations:** Any skin manifestation: *n* of events ST: *n* = 13 DE: *n* = 24 Serious skin manifestations: *n* of events ST: *n* = 0 DE: *n* = 2
Gratacós (Arthritis Res Ther, 2019) [38] (REDES-TNF)	RCT	123 (62 ST; 61 DE)	Axial spondylo-arthritis	Adalimumab Etanercept Golimumab Infliximab	Interval lengthening or dose reduction	12 months	**Occurrence of new infections:** ST: *n* = 19 (30.6%, 20.6–43.0) DE: *n* = 11 (18.0%, 10.4–29.5)
Atalay (JAMA Dermatol, 2020) [39] (CONDOR)	RCT	80 (41 ST; 39 DE, 40 patients that were treated with ustekinumab were excluded in this systematic review)	Psoriasis	Adalimumab Etanercept	Interval lengthening	12 months	**Occurrence of new infections: ^1^** ST: *n* = 26 (63.4%, 48.1–76.4) DE: *n* = 24 (61.5%, 45.9–75.1) **Occurrence of new skin manifestations: ^1^** ST: *n* = 6 (7.3%, 2.5–19.4) DE: *n* = 2 (5.1%, 1.4–16.9)
Landewé (Ann Rheum Dis, 2020) [40] (C-OPTIMISE)	RCT	209 (104 ST; 105 DE)	Axial spondylo-arthritis	Certolizumab pegol	Interval lengthening	48 weeks	**Occurrence of new infections:** Opportunistic infections: ST: *n* = 1 (1.0%, 0.2–5.2) DE: *n* = 3 (2.9%, 1.0–8.1) Oral candidiasis: ST: *n* = 0 (0.0%, 0.0–3.6) DE: *n* = 1 (1%, 0.2–5.2)
Emery (Ann Rheum Dis, 2020) [41] (PREDICTRA)	RCT	102 (39 ST; 102 DE, ST are patients receiving ST after failure of DE)	Rheumatoid arthritis	Adalimumab	Interval lengthening	ST: 16 weeks DE: 36 weeks	**Occurrence of new infections:** ST: *n* = 15 (38.5%, 24.9–54.1) DE: *n* = 34 (33.3%, 24.9–42.9)
Bertrand (Scand J Rheumatol, 2021) [42] (TapERA)	RCT	66 (34 ST; 32 DE)	Rheumatoid arthritis	Etanercept	Interval lengthening	1 year	**Occurrence of new infections: ^1^** Any infection: ST: *n* = 7 (20.6%, 10.4–36.8) DE: *n* = 7 (21.9%, 11.0–38.8) Serious infections: ST: *n* = 0 (0.0%, 0.0–10.7) DE: *n* = 0 (0.0%, 0.0–10.1)
Park (Clin Exp Rheumatol, 2016) [43]	Cohort study	83 (31 ST; 52 DE)	Axial spondylo-arthritis	Etanercept	Dose reduction	536.8 PYs (95.9 PYs ST; 440.9 PYs DE)	**Occurrence of new infections:** Any infection: incidence rate per 100 PYs (95% CI): ST: 17.7 (10.3–28.4) DE: 21.0 (17.0–25.8) Incidence rate ratio DE/ST: 1.194 (95% CI 0.712–2.002), *p* = 0.501 Clinically significant infections: incidence rate per 100 PYs (95% CI): ST: 1.0 (0.0–5.8) DE: 0.5 (0.0–1.6) Incidence rate ratio DE/ST: 0.435 (95% CI 0.039–4.795), *p* = 0.497 **Occurrence of new skin manifestations:** Any injection site reaction: incidence rate per 100 PYs (95% CI): ST: 8.3 (3.6–16.4) DE: 2.7 (1.4–4.8) Incidence rate ratio DE/ST: 0.327 (95% CI 0.134–0.801), *p* = 0.014 Clinically significant injection site reactions: incidence rate per 100 PYs (95% CI): ST: 1.0 (0.0–5.8) DE: 0.7 (0.1–2.0) Incidence rate ratio DE/ST: 0.652 (95 % CI 0.068–6.270), *p* = 0.711
Závada (Ann Rheum Dis, 2016) [44]	Cohort study	136 (83 ST; 53 DE)	Axial spondylo-arthritis	Adalimumab Etanercept Infliximab	Interval lengthening, dose reduction, or a combination	12 months	**Occurrence of new infections:** ST: 9 (10.8%, 5.8–19.3) DE: 4 (7.5%, 3.0–17.9) *p* = 0.550
Li (Arch Med Sci, 2019) [45]	Cohort study	96 (48 ST; 48 DE)	Axial spondylo-arthritis	Etanercept	Dose reduction	48 weeks	**Occurrence of new infections:** Upper respiratory tract infections: ST: *n* = 8 (16.7%, 8.7–29.6) DE: *n* = 6 (12.5%, 5.9–24.7) Tuberculosis infection: ST: *n* = 1 (2.08%, 0.4–10.9) DE: *n* = 1 (2.08%, 0.4–10.9) **Occurrence of new skin manifestations:** Injection site reactions: ST: *n* = 8 (16.7%, 8.7–29.6) DE: *n* = 5 (10.4%, 4.5–22.2)
Pouillon (Dig Liver Dis, 2019) [46]	Cohort study	56 (0 ST; 56 DE)	Inflammatory bowel disease	Adalimumab	Interval lengthening	Median of 15.9 months (IQR 7.9–30.6)	**Disappearance of skin manifestations:** *n* of patients in whom skin manifestations disappeared/*n* at baseline DE: *n* = 4/7 (57.1%, 25.1–84.2)
Atalay (J Dermatolog Treat, 2021) [47] (extension of CONDOR)	Cohort study	58 (40 ST; 18 DE, 30 patients that were treated with ustekinumab were excluded in this systematic review)	Psoriasis	Adalimumab	Interval lengthening	12 months	**Occurrence of new infections: ^1^** ST: 20 (50.0%, 35.2–64.8) DE: 11 (61.1%, 38.6–79.7) **Occurrence of new skin manifestations: ^1^** ST: 5 (12.5%, 5.5–26.1) DE: 0 (0.0%, 0.0–17.6)
van Steenbergen (Aliment Pharmacol Ther, 2017) [48]	Case–control study	80 (40 ST; 40 DE)	Inflammatory bowel disease	Adalimumab	Interval lengthening	Median of 37.1 months (IQR 30.2–49.3)	**Disappearance of skin manifestations:** *n* of patients in whom skin manifestations disappeared/*n* at baseline ST: 0/11 (0.0%) DE: 8/17 (47.1%, 26.2–69.0) **Disappearance of infections:** Frequent infectious symptoms: *n* of patients in whom frequent infectious symptoms disappeared/*n* at baseline ST: 0/5 (0.0%) DE: 5/7 (71.4%, 26.2–69.0)

CI—confidence interval; DE—de-escalation; IQR—interquartile range; PYs—person years; RCT—randomized controlled trial; SD—standard deviation; ST—standard treatment. ^1^ Requested data. ^2^ The DE group continued on the standard dosing interval for 4 weeks, followed by the lengthened interval for 8 weeks.

**Table 2 biomedicines-10-01034-t002:** Risk of bias of non-RCTs (Newcastle–Ottawa scale).

Study	Design	Selection (Max. 4 Stars)	Comparability (Max. 2 Stars)	Outcome */Exposure (Max. 3 Stars)	Total Stars (Quality)
Park (Clin Exp Rheumatol, 2016)	Cohort	★★★★	★★	★★	8 (good)
Závada (Ann Rheum Dis, 2016)	Cohort	★★★★	★★	★★	8 (good)
Li (Arch Med Sci, 2019)	Cohort	★★★★	★★	★★	8 (good)
Pouillon (Dig Liver Dis, 2019)	Cohort	★★★	-	★	4 (poor)
Atalay (J Dermatolog Treat, 2021)	Cohort	★★★★	★★	★	7 (poor)
van Steenbergen (Aliment Pharmacol Ther, 2017)	Case–control	★★★★	★	★★★	8 (good)

* Scored for outcome of interest of this systematic review (i.e., the occurrence of new infections and/or skin manifestations, or the reduction and/or disappearance of infections and/or skin manifestations).

## Data Availability

Not applicable.

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
