# Peer review of "De-Escalation of Anti-Tumor Necrosis Factor Alpha Agents and Reduction in Adverse Effects: A Systematic Review"

_biomedicines, 2022, doi:10.3390/biomedicines10051034_

Round 1
Reviewer 1 Report
- From the abstract I have the impression that we are studying the article, and not the content of the article!
- "Currently, there is insufficient evidence to conclude ..." That is not a conclusion. From the little evidence, please draw a conclusion!
- "Our findings are consistent with a meta-analysis performed by Vinson et al. .. - Is there anything new in your article?
Conclusion: Please read the article, and present the content, the conclusion of each article, not the quality of the articles found.
- "In conclusion, the available evidence is insufficient to either refute or support deescalation as a means to reduce infections or skin manifestations in patients on standard-dosed anti-TNF treatment." - You studied 2706 patients. I don't think there is enough evidence! Any study involving at least 100 patients is considered statistically significant. You have studied 2706 patients.
Author Response
Please see the attachment (page 2-3)

Reviewer 2 Report
Anti TNF agents are important group of clinically used agents with significant share adverse effect and clinical failure. Systematic review with this topic can be interesting for scientist and medics and significantly participate to better understanding of this problematics. I especially appreciate that systematic and careful and detailed authors approach, which allows easy orientation and acquaintance with discussed topic. Nevertheless, some point can be taken for the improvement of manuscript.
Authors should be also included chapter, subchapter focused on the clinical failure.
For the better illustration of Anti TNF agents and their therapeutic mechanism, authors should include figure, or schema.
Minor
Line 35 Used abbreviation of tumor necrosis factor alpha is TNF-alfa, not only TNF.
Author Response
Please see the attachment (page 4)

Round 2
Reviewer 1 Report
1."The studies described patients with axial spondyloarthritis (n=8), 20
rheumatoid arthritis (n=7), psoriasis (n=3), or inflammatory bowel disease (n=2). De-escalation strategies included interval lengthening (n=12), dose reduction (n=6) or both (n=2)."
- Instead of n, please show in parentheses the number of patients in those studies.
2."but the level of evidence is low. Currently, there is insufficient evidence to conclude
whether dose de-escalation of anti-TNF reduces infections or skin reactions. A de-escalation strategy should therefore not be recommended for the sole purpose of reducing drug-related adverse effects. Meticulous documentation of adverse effects is recommended to further address this question. Registration: PROSPERO CRD42021252977."
- Please delete this. I maintain my statement. You studied 2706 patients. From the study of these patients you have to draw a conclusion. Not to say that there is little evidence to support this!
Author Response
1."The studies described patients with axial spondyloarthritis (n=8), 20
rheumatoid arthritis (n=7), psoriasis (n=3), or inflammatory bowel disease (n=2). De-escalation strategies included interval lengthening (n=12), dose reduction (n=6) or both (n=2)."
Instead of n, please show in parentheses the number of patients in those studies.
Response: We have added the number of patients to the abstract.
2."but the level of evidence is low. Currently, there is insufficient evidence to conclude
whether dose de-escalation of anti-TNF reduces infections or skin reactions. A de-escalation strategy should therefore not be recommended for the sole purpose of reducing drug-related adverse effects. Meticulous documentation of adverse effects is recommended to further address this question. Registration: PROSPERO CRD42021252977."
Please delete this. I maintain my statement. You studied 2706 patients. From the study of these patients you have to draw a conclusion. Not to say that there is little evidence to support this!
Response: We have adapted this in the manuscript.